# Facilitators of and barriers to healthcare providers' adoption of harm reduction in cannabis use: a scoping review protocol

Roula Haddad ,[1] Christian Dagenais,[1] Christophe Huynh,[2,3] Jean-Sébastien Fallu[2,4,5]

For numbered affiliations see end of article.

**Correspondence to**
Roula Haddad;
roula.haddad@umontreal.ca

## ABSTRACT

**Introduction** The high prevalence of cannabis use and the potential for negative effects indicate the need for effective prevention strategies and treatment of people who use cannabis. Studies show that harm reduction (HR) in cannabis use is effective in minimising the harmful consequences of the substance. However, health professionals often misunderstand it and resist its adoption due to various obstacles. To our knowledge, there has been no review of the scientific literature on the factors that facilitate or hinder practitioners' adoption of HR in cannabis use. To fill this gap, we aim to identify, through a scoping review, facilitators and barriers to healthcare providers' adoption of HR in cannabis use in Organisation for Economic Cooperation and Development (OECD) countries.

**Methods and analysis** Our methodology will be guided by the six-step model initially proposed by Arksey and O'Malley (2005). The search strategy will be executed on different databases (Medline, PsycINFO, CINAHL, Web of Science, Embase, Sociological Abstracts, Érudit, BASE, Google Web and Google Scholar) and will cover articles published between 1990 and October 2022. Empirical studies published in French or English in an OECD country and identifying factors that facilitate or hinder healthcare providers' adoption of HR in cannabis use, will be included. Reference lists of the selected articles as well as relevant systematic reviews will be scanned to identify any missed publications by the electronic searches.

**Ethics and dissemination** Ethics approval is not required. The results will be disseminated through various activities (eg, publication in peer-reviewed journals, conferences, webinars and knowledge translation activities). The results will also allow us to conduct a future study aiming to develop and implement a knowledge translation process among healthcare practitioners working with youth in Quebec in order to enhance their adoption of HR in cannabis use.

## STRENGTHS AND LIMITATIONS OF THIS STUDY

⇒ The search strategy was codeveloped by two information management specialists in the addiction and knowledge translation fields.
⇒ The search strategy will aim to retrieve published articles on several health databases and unpublished studies found in the grey literature.
⇒ Two reviewers will independently select the studies to be included in the scoping review throughout the entire study selection stage.
⇒ The included studies will be limited to those published in French or English.
⇒ The included studies will be limited to those published in Organisation for Economic Cooperation and Development countries.

## INTRODUCTION

### Cannabis use

Psychoactive substances (eg, cannabis, alcohol, nicotine) are defined as substances whose use affects mental processes (eg, perception, cognition, emotions, mood) and behaviours without necessarily leading to addiction.[1] After tobacco and alcohol, cannabis represents the third most consumed psychoactive substance globally among adults and youth.[2–5] Canada remains among the developed countries with the highest rates of cannabis use among young people and adults, with high prevalence in Quebec and multiple patterns of use (eg, smoking, eating, vaporising, vaping).[2 5–9] In 2023, 26% of those 16 years of age and older reported using cannabis in the 12 months prior to the study, with a higher prevalence among those aged between 20 and 24 years old.[10]

The lifetime cumulative probability of transitioning from use to dependence was found to be the lowest for people who use cannabis (PWUC) (8.9%) compared with other substances such as nicotine (67.5%), alcohol (22.7%) and cocaine (20.9%); this can be due to several factors such as infrequency in use, consumption of cannabis with low rates of δ-9-tetrahydrocannabinol, availability, legality and social acceptability.[9 11–15] Even with long-term exposure to cannabis, PWUC

do not necessarily develop severe problems or a cannabis dependence.[5 15 16]

However, PWUC intensively or at-risk populations (ie, pregnant persons, people presenting respiratory problems, a mental health comorbidity, having an early onset of continued tobacco use or concurrently smoking tobacco and cannabis) may develop a cannabis use disorder or experience harm at several levels.[5 12 16–19] These harms may include a decreased academic or professional performance, a cognitive impairment, a deterioration of mental health (eg, development of psychosis or depression), an increased occurrence of risky behaviours (eg, cannabis-impaired driving), etc.[3–5 12 14 18 20–22] It is important to specify that a severe cannabis-induced mental health condition (eg, psychosis) might occur among only 2% of PWUC.[5] Despite the minimal probability of leading to potentially serious adverse consequences, the high prevalence of cannabis use as well as the potential harms that might be experienced, make it essential to implement effective intervention programmes among PWUC.[7 12 15 17]

## Abstinence approach
To address this reality, prevention and treatment programmes based on the abstinence approach (ie, total elimination of cannabis use), have been widely implemented.[11 13] The emergence of these programmes has been also influenced by policies such as the 'War on Drugs'.[23] The abstinence-based model forms the basis of many programmes developed to prevent or treat problematic cannabis use and has been applied with at-risk or marginalised populations (eg, youth in the foster care system).[21 24] Despite its potential to decrease the frequency or amount of substance use, the abstinence approach presents limited evidence to support its effectiveness and has been criticised for various reasons.[21 24–26] First, it does not provide PWUC with the necessary skills to identify and mitigate the harms associated with their use.[21 27] Second, the abstinence approach tends to focus more on the negative consequences of use through strategies that evoke fear, without necessarily taking into consideration the social context of cannabis use.[20 21] Third, the risks of relapse and dropout in these programmes are also found to be high, leaving a significant number of PWUC for whom this goal remains unattainable.[20 28] Given these limitations of abstinence-oriented programmes, other alternative and more flexible treatments, such as harm reduction (HR), are essential to reduce and mitigate cannabis-related harms.[14 21 28 29]

## Harm reduction
### Description
HR in cannabis use aims to minimise the harmful consequences of the substance at the individual, psychological, legal and social levels among PWUC.[19 20 28 29] It offers a public health framework based on values of pragmatism and humanism, as it does not view substance use through a moral lens, but as an inevitable societal fact of long-standing.[4 11 30 31] Whether among adults or adolescents,

elimination of substance use is unrealistic at the population level and should be the individual's choice without being imposed, as it represents an unwanted and impractical goal for some (eg, in case of recreational or occasional use, in case of dual diagnosis combining psychiatric and substance use disorders).[27–29 32 33] HR in cannabis use also seeks to equip PWUC to make responsible and rational decisions and learn ways to reduce the negative consequences associated with their consumption.[11 12 20 27 29 31] To this end, HR clarifies the notion of safe substance use that is determined by the interaction of three components: the individual (height, weight, gender, physical and mental health status, state of mind, etc); the drug (quantity, frequency of use, tolerance to the product, combination with other products, quality, etc); and the setting (location, time of day, interpersonal relationships, conflicts, laws, etc).[4 20 27] In addition, HR takes into account the personal characteristics of PWUC (impulsivity, sensation-seeking, etc) and addresses their potential ambivalence about stopping substance use, their feelings of failure on relapse, their engagement in treatment, their social skills, their emotional regulation, etc.[20] In the 'Lower Risk Cannabis Use Guidelines', Fischer *et al*[5] updated the initial recommendations to reduce the harms of cannabis use. These recommendations include delaying the initiation of cannabis use until late adolescence or the completion of puberty, consuming low-potency cannabis products, avoiding deep inhalation, using legal and quality-controlled cannabis products, etc.[5]

### HR effectiveness
Interventions based on HR for non-injected drugs have been studied across various populations (eg, youth, adults, people in housing programmes presenting mental health conditions) and have shown promising results in decreasing the negative consequences associated with substance use.[20 28]

Several studies showed the effectiveness of HR strategies and current school-based HR programmes (eg, SHAHRP in the UK and SCIDUA in Canada) in developing safer attitudes toward substance use and in reducing negative consequences related to use.[21 22 34 35] The effectiveness of HR among youth who use cannabis has led the University Institute on Addictions (*Institut universitaire sur les dépendances*) in Quebec to recommend it as an intervention modality among this clientele.[36] In addition, effective early interventions targeting college and university students with at-risk cannabis use are those that reduce the harms associated with cannabis use.[17 36–38] Moreover, Palfai *et al*[38] found that students participating in a web-based HR intervention (Marijuana eCHECKUP TO GO) showed statistically significant lower results in peer cannabis use after 6 months ($f^2$=0.11 (B=7.45 (3.34), p<0.05)).

HR in cannabis use was also found effective among adults.[17 39] After delivering a brief HR intervention for PWUC, Fischer *et al*[17] found significant reductions in risk outcome indicators only among the experimental group.

At the 12-month follow-up, a change was maintained for 'deep inhalation/breath-holding' (experimental group: Q=13.1; $p<0.05$; control group: Q=4.8; $p>0.05$), and 'driving after cannabis use' (experimental group: Q=9.3; $p<0.05$; control group: Q=0.9; $p>0.05$).[17] Furthermore, without completely abstaining from the substance, a functional improvement can be reached by treatment-seeking adults presenting a cannabis use disorder when they reduce the frequency and/or quantity of cannabis use.[39] Reduction in the frequency of cannabis use was associated with a decrease in depression (F=2.76, p=0.04, $n_p^2$=0.04), anxiety (F=3.70, p=0.01, $n_p^2$=0.05) and cannabis-related problems (F=8.95, p<0.001, $n_p^2$=0.12).[39] In addition, a decrease in the quantity of cannabis consumption was associated with a decrease in anxiety (F=3.02, p=0.03, $n_p^2$=0.04) and cannabis-related problems (F=3.24, p=0.02, $n_p^2$=0.05).[39] A systematic review also highlighted that the adoption of HR strategies by PWUC acts as a protective factor for people with poor mental health, low self-regulation, high impulsivity and high negative urgency.[22]

## Acceptability of HR

Despite its proven effectiveness, the acceptability and applicability of HR by health and social services practitioners remain limited, and various factors facilitate or hinder its adoption.[21 40–42] MacCoun[43] showed that, among practitioners who did not adopt HR, some had based their decision on moral grounds, regardless of its effectiveness. Various barriers limit its use by practitioners. First, ambiguities in its conceptualisation play an influential role; for example, some practitioners perceive HR as sending the wrong message, that is, one of tolerating or even encouraging substance use.[27 28 30 44] Some do not perceive total cessation of substance use as a legitimate goal that could be achieved through HR.[45] Also, there is often confusion between reducing use (frequency, quantity, etc) and reducing harm (modifying consumption practices, such as contexts and mixtures, to reduce consequences).[30] These misconceptions point to the need for awareness-raising, training and supervision of practitioners interested in this approach.[28] Second, the adoption of HR can be hindered by ethical dilemmas, as well as by issues related to the personal and collective values of healthcare workers and the therapeutic model of abstinence.[30] Indeed, it runs counter to traditional treatments by tolerating risky behaviours and accepting that HR in drug use is a legitimate outcome.[30 44] Practitioners may also fear the emergence of legal, social and health problems among their clients.[28] Third, its adoption may be limited by contextual barriers, such as lack of funding, stigma that undermines demand for care, resistance from local jurisdictions, and lack of services and trained personnel, particularly in the mental health sector.[41] Healthcare providers' resistance to applying HR in cannabis use leads to limited knowledge and utilisation of effective HR techniques and guidelines among PWUC.[5 25] Among the study's participants (ie, PWUC), Kruger et al[25] found that less than half of the participants

believed that the listed HR techniques were effective and reported applying them.

However, several factors seen as HR benefits have been found to facilitate its implementation by healthcare providers, such as: broadening the spectrum of acceptable goals, improving clients' decision-making skills, creating positive and quality relationships, and managing relapses.[28] A study by Sharp et al[41] showed that clarifying the positive impacts of HR at the community level (eg, safety) and ensuring the availability of resources could increase the likelihood of its adoption.

## Purpose

Despite its proven effectiveness, several reasons might hinder HR adoption by health professionals among PWUC.[30 41] However, to date, there has been no review of the scientific literature that identifies the factors that facilitate or limit the adoption of HR in cannabis use. To fill this gap, we aim to identify, through a scoping review, facilitators and barriers to healthcare providers' adoption of HR in cannabis use.

## METHODS AND ANALYSIS

This study will follow the methodological steps of scoping reviews.[46] This type of review has become more prevalent in recent years and is a type of knowledge synthesis review.[46 47] There is no universal definition for scoping reviews; however, a variety of factors distinguish them from other types of knowledge synthesis.[48] First, scoping reviews address broad research questions and include studies with different designs and multiple sources of evidence to provide an overview of the available knowledge around a concept.[49] On the contrary, a systematic review following Cochrane standards explores more specific research questions based on detailed inclusion and exclusion criteria.[50–52] Second, while assessment of the methodological quality of included studies is recommended for scoping reviews, it is not mandatory, whereas assessment of the risk of bias of included studies is required for Cochrane-type systematic reviews.[50 52] Researchers undertake scoping reviews for a variety of reasons: (1) to review research activity in a given area; (2) to determine the feasibility and appropriateness of conducting a systematic review based on Cochrane standards; (3) to summarise and disseminate the results of existing research on a topic; and/or (4) to identify a gap in the literature and draw conclusions regarding a topic.[49] Our methodological choice is underpinned by three of these reasons: once the research activity around the topic has been consulted, the findings will be summarised and used to support a second study aimed at disseminating knowledge to practitioners through a knowledge translation process. This will also allow us to identify gaps in the literature and draw conclusions related to the topic.

Arksey and O'Malley[49] were the first to propose a six-step model for conducting scoping reviews. Our methodology will be guided by this model, which was later

refined by Levac et al[18] and revised by members of the Joanna Briggs Institute.[50] The six stages we will follow are:

► Stage 1: determining the research question and the objective.
► Stage 2: identifying relevant studies.
► Stage 3: selecting studies.
► Stage 4: charting the data.
► Stage 5: collating, summarising and reporting the results.
► Stage 6: conducting a consultation exercise.

The research protocol will be reported using the Preferred Reporting Items for Systematic Reviews and Meta-Analyses (PRISMA) Extension for Scoping Reviews grid.[47] This grid is an extension of the PRISMA grid originally developed for Cochrane-type systematic reviews and helps to ensure the transparency and reproducibility of the study.[47 50]

## Stage 1: determining the research question and the objective

Arksey and O'Malley's[49] model suggests that scoping reviews should begin not only with identifying research questions but also with clarifying the resulting objectives.[48 50] Our scoping review is exploratory and aims to identify facilitators and barriers to healthcare providers' adoption of HR in cannabis use in Organisation for Economic Cooperation and Development (OECD) countries. Based on the Population–Concept–Context (PCC) model, which allows the broad scope of the study to be respected without specifying restrictive inclusion criteria,[50] we formulated the research question: what factors influence providers (population) in the healthcare field (context) to adopt HR in cannabis use (concept)? Specific research questions associated with the components of the PCC model were also identified.

► Question 1, related to the concept and context components: what are the facilitators and barriers to healthcare providers' adoption of HR in cannabis use?
► Question 2, related to the population component: who is the clientele of the providers identified in the studies?
► Question 3, related to the concept component: what is the definition of HR in cannabis use?

## Stage 2: identifying relevant studies
### Search strategy
The search strategy was developed through an iterative process. A senior librarian at the Quebec Addiction Library (*Bibliothèque québécoise sur les dépendances*) first developed three search strategies with different concepts and ran them on the Medline database. The first 50 results of each strategy were consulted, which led us to opt the one that grouped key terms related to the following concepts: HR, clinicians, and cannabis (see online supplemental appendix). The search strategy was then reviewed by a second information professional working in the RENARD Team for Knowledge Translation who, in turn, adapted it to the selected databases. The final search strategy executed on all the databases was then validated by the RENARD Team information specialist. The Peer Review of Electronic Search Strategies tool served as a guide for the librarians in this process.[53] The search strategy executed on each database is presented in online supplemental appendix. All search strategies were executed on 10 October 2022.

### Information sources
To identify relevant published and unpublished studies for inclusion in the study, various sources of information will be reviewed.[46 49 50] With the guidance of the two librarians, the search strategy will be executed on the leading health and intervention databases: Medline, PsycINFO, Cumulative Index to Nursing and Allied Health Literature (CINAHL), Web of Science, Embase and Sociological Abstracts. To explore the grey literature (eg, theses, research reports, etc), the search strategy will be adapted to the Google Web and Google Scholar search engines, as well as the Érudit (French database) and BASE databases. All documents identified will be entered into Zotero software for the research team members to access. To identify any missed publication by the electronic searches, reference lists of selected articles will be manually searched and relevant systematic reviews will be scanned to identify their included studies.

## Stage 3: study selection
After running the search strategy on the selected databases and completing the second stage, all identified duplicates will be removed. The remaining documents will then be entered into Covidence software. Two reviewers will independently select relevant documents for inclusion by reading the titles and abstracts of the identified studies. Their decisions will be based on the specified inclusion and exclusion criteria. The inter-rater agreement between the reviewers will first be calculated and then they will meet regularly to resolve selection conflicts and refine the eligibility criteria as needed. After this first step, the documents selected and deemed potentially relevant will then be the subject of the second step, the full-text reading. Again, the two reviewers will independently record their choices on Covidence. The inter-rater agreement for this step will be calculated and the reviewers will then resolve any new conflicts. A third reviewer will be called on as needed for any conflict resolution. This will complete the document selection stage, whose steps will be presented in a PRISMA diagram.[54]

### Inclusion and exclusion criteria
Inclusion and exclusion criteria have been specified and will be fine-tuned as needed to select relevant studies (table 1). Using this process, empirical studies of quantitative, qualitative or mixed designs, identifying factors that facilitate or hinder healthcare providers' adoption of HR in cannabis use, will be selected. To facilitate comparison and generalisation of the results to the Quebec context, the review will be limited to studies conducted in any of the 38 OECD countries. Articles published between 1990 and

**Table 1** Inclusion and exclusion criteria

| Criteria | Inclusion criteria | Exclusion criteria |
|---|---|---|
| Type of study | ▶ Empirical study: quantitative, qualitative or mixed | ▶ Study that does not present empirical results (eg, theoretical study, conceptual framework, etc) or knowledge review (eg, systematic or literature review)<br>▶ Interviews |
| Type of documents | ▶ Peer-reviewed scientific articles, research reports, dissertations, theses | ▶ Books and practice guides |
| Conceptual framework | ▶ Harm reduction (HR) in cannabis use<br>▶ Cannabis risk reduction<br>▶ Non-abstinence in cannabis use | ▶ Another conceptual framework |
| Objective | ▶ Identification of factors* facilitating or hindering practitioners' adoption of the HR approach† in cannabis use | ▶ Evaluation of the effectiveness of interventions based on HR<br>OR<br>▶ Stakeholder perceptions of the use of cannabis as an HR strategy to circumvent the effects of other drugs<br>OR<br>▶ Attitudes toward decriminalisation of cannabis |
| Psychoactive substance being studied | ▶ Marijuana, hashish or cannabis for non-medical purposes<br>▶ 'Drug' if cannabis is part of its conceptualisation in the study | ▶ Any substance other than marijuana, hashish or non-medical cannabis (eg, tobacco, alcohol, medical cannabis, MDMA, Ecstasy)<br>▶ Study that focuses on 'performance and image enhancing drugs' or 'crack' or 'new psychoactive substances' |
| Target population | ▶ Practitioners‡ working in the health field<br>▶ Practitioners in training | ▶ PWUC§ |
| Country of study | ▶ Organisation for Economic Cooperation and Development countries | ▶ Other countries |
| Publication date | ▶ From 1990 onwards | ▶ Before 1990 |
| Language | ▶ French and/or English | ▶ Languages other than French or English or text not available |

*'Factors' include perceptions, beliefs, facilitators, obstacles, oppositions, attitudes, opinions, barriers, biases, motivations, preferences, determinants, incentives, influences and perspectives on the adoption of HR in cannabis use, as well as its acceptability and receptivity.
†'Approach' refers to strategies, interventions, practices, services, methods, techniques, treatments, programmes or guides for the HR approach in cannabis use.
‡'Practitioners' include healthcare personnel, professionals, or practitioners, allied healthcare personnel, professionals, or practitioners, social workers, counsellors, psychoeducators, educators, nurses, criminologists, psychologists, clinicians, caregivers, therapists, psychotherapists and physicians.
§Studies addressing the views of people who use cannabis (PWUC) regarding HR or its adoption by practitioners will be excluded.

October 2022 will be included, as it was early in the 1990s that HR gained international prominence and its scope of application began to expand. Papers not meeting these inclusion criteria will be excluded. Systematic reviews will be excluded to avoid duplication and ensure equal representation of the selected papers; the executed search strategy might have already captured studies included in a potentially relevant systematic review. However, systematic reviews' reference lists will be examined to identify additional relevant studies that might be selected.

### Stage 4: charting the data
To analyse the selected studies on a common basis, specific variables of interest will be identified based on the research questions.[49] These will form the components of summary sheets that will be developed in Microsoft Excel and used to extract results (table 2). This method is an analytical descriptive recording of the data.[49 50 55] The first author (RH) will extract the data from the included studies and create the summary sheets. The research supervisor (CD) will validate the summary sheets throughout the process and ensure their alignment with the research questions.[50]

### Stage 5: collating, summarising and reporting the results
Based on the eligibility criteria, studies deemed relevant will be collected, summarised and reported. They will be subjected to (1) a numerical analysis and (2) a narrative organisation encompassing a descriptive qualitative

**Table 2** Summary sheets

| General variables | Specific variables |
|---|---|
| General characteristics of the study | Study title<br>Author(s)<br>Language of publication<br>Date of publication<br>Period of publication<br>Journal<br>Type of article<br>Full reference<br>Country of study<br>Psychoactive substance under study<br>The legal status of cannabis in the country of study |
| Introduction | Main concepts<br>Definition of the main concept: harm reduction (HR) in cannabis use<br>Research question(s)<br>Objective(s)<br>Hypothesis |
| Methodology | Study design<br>Target population<br>Place of work of the target population<br>Inclusion criteria of participants<br>Recruitment method<br>Sample size<br>Country of origin of participants<br>The clientele of the population recruited<br>Data collection method<br>Analysis steps |
| Results | Sample presentation<br>Key findings: (1) facilitators and (2) barriers to practitioners' adoption of HR in cannabis use<br>Secondary outcomes or other results |
| Conclusion | Study strengths<br>Study limitations<br>Gaps in the literature and future research needs |

**Table 3** Narrative organisation of the included studies

| Data | Study 1 | Study 2 | Study … |
|---|---|---|---|
| Type of publication | | | |
| Date of publication | | | |
| Country of study | | | |
| Legal status of cannabis in the country of the study | | | |
| Definition of harm reduction to cannabis use | | | |
| Design of the study | | | |
| Target population | | | |
| Place of work of the target population | | | |
| The clientele of the target population | | | |
| Data collection method | | | |
| Key findings | | | |
| Facilitators or enabling conditions | | | |
| Barriers or adverse conditions | | | |
| Secondary outcomes | | | |

analysis.[47 49 50] A numerical analysis of the scope, nature and distribution of the included studies will be performed to various characteristics: date of publication, country of origin of the studies and type of document. Subsequently, a narrative organisation of the results will be produced to identify the relationships between the data and the research questions. The summary sheets will be combined, tabulated and synthesised, and will then be subjected to a descriptive qualitative analysis (table 3).

## Stage 6: consultation exercise

Expert consultation is an optional step that promotes methodological rigour in scoping reviews.[49] In this study, the project's supervisor and coresearchers will be solicited as consultants. Members of the RENARD team and researchers involved in the field of substance use and HR will be consulted to help clarify findings and validate the resulting recommendations.[49] Consultations will be conducted: (1) after preliminary results have been obtained and (2) after analyses of the results have been completed.

## Patient and public involvement

None.

## Ethics and dissemination

To our knowledge, this is the first scoping review on factors that facilitate or hinder healthcare providers' adoption of HR in cannabis use. Other reviews have studied HR interventions in general among practitioners working with a specific population. This study will provide a clear picture of the factors at play when adopting HR, and the results could potentially be generalisable to OECD countries. The present study is exempt from ethics approval because it involves no patient or personal data collection. The results are expected to be ready by March 2024. They will be disseminated, alongside the scoping review protocol, through various activities (eg, publication in peer-reviewed journals, conferences, webinars, posters, *Three Minute Thesis* competition).

After completing the scoping review, we will be able to conduct a future study aiming to implement a knowledge translation plan among practitioners working with youth in Quebec to enhance and expand their adoption of HR in cannabis use.

**Author affiliations**
[1]Department of Psychology, Université de Montréal, Montreal, Quebec, Canada
[2]University Institute on Addictions, Centre intégré universitaire de santé et de services sociaux du Centre-Sud-de-l'Île-de-Montréal, Montreal, Quebec, Canada
[3]Department of Psychiatry, Université de Montréal, Montreal, Quebec, Canada
[4]School of Psychoeducation, Université de Montréal, Montreal, Quebec, Canada
[5]Centre de recherche en santé publique (CReSP), Centre intégré universitaire de santé et de services sociaux du Centre-Sud-de-l'Île-de-Montréal, Montreal, Quebec, Canada

**Acknowledgements** We wish to thank Karine Bélanger, senior librarian at the Bibliothèque québécoise sur les dépendances in Quebec for developing the initial search strategy used in this scoping review. We also wish to thank Julie Desnoyers, the information specialist affiliated with the RENARD Team for Knowledge Translation at Université de Montréal, for validating the search strategy and adapting it to the selected databases.

**Contributors** RH, CD, CH and J-SF conceptualised the study. RH drafted the protocol. CD, CH and J-SF critically revised the manuscript. RH wrote the final draft manuscript and all the authors approved it.

**Funding** This work was supported by the Ministère de la Santé et des Services sociaux (MSSS), Fonds de recherche du Québec – Santé (FRQS), and Fonds de recherche du Québec – Société et culture (FRQSC) (Grant number: 2023-0PTR-322652). A grant obtained by the Centre de recherche en santé publique (CReSP) helped in partially covering the revision and publication fees of this scoping review protocol.

**Competing interests** None declared.

**Patient and public involvement** Patients and/or the public were not involved in the design, or conduct, or reporting, or dissemination plans of this research.

**Patient consent for publication** Not applicable.

**Provenance and peer review** Not commissioned; externally peer reviewed.

**ORCID iD**
Roula Haddad http://orcid.org/0000-0002-0414-7144

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
