## [Reviewer comments · BMJ Open]

ARTICLE DETAILS

TITLE (PROVISIONAL)	Facilitators of and barriers to healthcare providers' adoption of harm reduction in cannabis use: a scoping review protocol
AUTHORS	Haddad, Roula; Dagenais, Christian; Huynh, Christophe; Fallu, Jean-Sébastien

VERSION 1 – REVIEW

REVIEWER	Grossman, Ellie Cambridge Health Alliance
REVIEW RETURNED	24-Jan-2024

GENERAL COMMENTS	Review of 'Facilitators and barriers to healthcare providers' adoption of the harm reduction approach to cannabis use: a scoping review protocol' Summary thought: The authors present a protocol paper describing a plan for a scoping review. They plan to explore evidence for barriers and facilitators of health-care providers' adopting a harm reduction approach to cannabis use. They plan to use this information to implement programming among staff at addiction treatment facilities in Quebec, with a focus on programs for adolescents and young adults – although they choose not to restrict their scoping review to this specific age range. Major Essential Revisions: • The Introduction section on 'HR efficacy' would be more compelling if the authors added quantitative data supporting effectiveness and/or efficacy of harm-reduction programs. Minor Essential Revisions: • The authors should include the specific dates for their literature search.• It is confusing (and contradicts their thesis) that the authors say the rate of cannabis use in Canada and Quebec is declining (Introduction, first paragraph) – because it seems to contradict data from other sources and parts of the world, and it weakens the impetus for the authors' study. I would suggest the authors omit this sentence.• One aim of the authors' study is to provide evidence they can use in improving systems of care for adolescents and young adults. In the Introduction section, some of their rationale and background focuses on this population as well (e.g. the 'HR efficacy' section). However, their scoping review does not restrict inclusion based on
---

	targeted population age – which makes for a mismatch between background/rationale provided and the actual study planned. The manuscript may hang together better if the authors focus on a non-age-restricted rationale for their study, and then mention a focus on providers serving adolescents and young adults as an exploratory substudy (if there is sufficient existing literature found). Other Comments:  • A strength of this study is its use of the term ‘healthcare providers’; I would expect that the authors will find studies involving a range of different types of health-care workers. These professional differences may need to be taken into account in the analytic plan or way of presenting results. • The authors do a nice job of explaining what a scoping review is (first paragraph of Methods/Analysis section). Table 1 is also very useful.
--	---

REVIEWER	Sohler, Nancy City University of New York, Csom
REVIEW RETURNED	24-Jan-2024

GENERAL COMMENTS	I think this is a nicely presented protocol. My suggestions are minor.  1. page 6: Efficacy and effectiveness are both used, although I think the studies are effectiveness studies. 2. The introductory sections seem to focus on young people, but this isn't highlighted in the main research question(s) and/or the methods. 3. I assume the years for the journal index searches are the same as for the database searches--is that correct? Also a list of journals to be included in this process probably should be identified ahead of time. 4. I'm confused about the use of inter-rate reliability in the article selection process. It seems the two reviewers will discuss disagreements and then reliability will be calculated. Wouldn't discussion lead to a consensus? 5. It appears that one person will be collecting all of the data after the studies are selected (with some validation from another person). I believe there is potential for bias and therefore >1 person could be included.
---

REVIEWER	Walsh, Hannah King's College London, Department of Addictions
REVIEW RETURNED	06-Feb-2024

GENERAL COMMENTS	Thank you for the opportunity to review this interesting and timely manuscript. Given the changing legal landscape and increasing use and availability of cannabis, it is important to consider harm reduction approaches within healthcare services, and other professionals or organisations which may provide support to young adults. I have a number of suggestions regarding the framing of the review, and the evidence based used to justify the review. I suggest this would benefit from a review and re-drafting, and the language is mostly clear but would benefit from a thorough proof-reading for
---

English, e.g. use of phrase 'psychiatric disabilities' indicates there may be a need for this.

The methodology appears sound and is clearly described, so I have only a couple of comments about that.

Overall, I think the paper would benefit from a clearer focus on cannabis harm reduction literature, and young adult harm reduction approaches. Currently, there are a number of references to the harm reduction approach in general, which arose from concerns primarily around opiate use in particular unsafe injecting; harm reduction approaches toward cannabis use are of course very different since the drug is very different, and I think this paper would benefit from outlining this, then focussing on cannabis harm reduction literature.

There are a number of key references missing which I have noted in the text below, and in general it would be good to see more recent references used, not least because the nature of cannabis including potency has changed so significantly in recent years.

I have noted down specific points in the text below:

1) P4 line 24 sentence

Other research suggests cannabis use transition to dependence is significant – not sure what 'limited addictive power' refers to but suggest rephrasing, also references cited there are old – suggest finding more recent references – e.g.

Lopez-Quintero, Catalina, et al. "Probability and predictors of transition from first use to dependence on nicotine, alcohol, cannabis, and cocaine: Results of the National Epidemiologic Survey on Alcohol and Related Conditions (NESARC)." *Drug and alcohol dependence* 115.1-2 (2011): 120-130.

Englund, Amir, et al. "Can we make cannabis safer?." *The Lancet Psychiatry* 4.8 (2017): 643-648.

2) Suggest rephrasing p4 line 28 – to discuss 'health and psychosocial harms and consequences'? As a general point, it might be useful to situate the issue more clearly within the context of harms and widespread use; since one of the issues with cannabis is that it is widely used and has negative and harmful consequences for a proportion of users – not all - but that vulnerability to harmful consequences may not be obvious prior to use, or may not prompt different choices about use. Some of the consequences can be very significant (e.g. psychosis) – so the frequency might be low but the impact is huge, hence harm reduction has an important role to play. Not least, co-administered tobacco use is a significant factor and may add to dependence risk. In this section, the literature cited is quite old – cannabis potency AND research has developed a lot in last ten yrs so it is important to reflect this

3) P4 line 38

Not all programs will have been based on abstinence – important to qualify language – eg. War on drugs etc has influenced aims of many prevention/treatment programs. Suggest including wider, recent references here. For example, the paper below is a key text: Fischer, Benedikt, et al. "Lower-Risk Cannabis Use Guidelines (LRCUG) for reducing health harms from non-medical cannabis use:

	A comprehensive evidence and recommendations update." International journal of drug policy 99 (2022): 103381. Overall, it might be easier to make the argument for this review by describing the literature focussing on harm reduction relating to cannabis only, rather than harm reduction per se. Harm reduction across all substance use is very broad, and as described already cannabis is a very different substance to heroin, cocaine use, which have driven the impetus for harm reduction approaches thus far. Therefore, it may be more useful to review extant literature on young adult harm reduction, and cannabis harm reduction as two foci. e.g. : Kruger, Jessica S., Daniel Kruger, and R. Lorraine Collins. "Knowledge and practice of harm reduction strategies among people who report frequent cannabis use." Health promotion practice 22.1 (2021): 24-30. Sherman, Brian J., et al. "Evaluating cannabis use risk reduction as an alternative clinical outcome for cannabis use disorder." Psychology of Addictive Behaviors 36.5 (2022): 505. Chatham, Carrington L. Harm Reduction: Social Workers' Attitudes toward Marijuana Use in Substance Abuse Treatment. Diss. Capella University, 2021. Winer, James Michael, et al. "Addressing adolescent substance use with a public health prevention framework: the case for harm reduction." Annals of medicine 54.1 (2022): 2123-2136. 4) L54 I'm not sure this is a strong argument – again it's difficult to make general statements about 'abstinence programs' – there may be plenty which do reference peer pressure and social context. The third argument is stronger. 5) P5 I26 – clarify that authors are describing the argument that HR is based upon – it reads like authors' own opinion 6) P6 I8 – for 'psychiatric disabilities' I assume you mean 'mental health conditions' ? 7) P6 I36 I suggest authors clarify which healthcare providers you are referring to (this is a huge group – are you focussing on substance use providers? Mental health?) –the statement that HR is 'controversial' – this seems a little strong. I suggest instead something like 'not widely accepted'. 8) P7 I 29 This statement needs qualifying – to justify that it is 'under-implemented' would require substantial references which have examined the adoption of HR approaches, for example at a policy level; (e.g. Emily J., et al. "Canadian cannabis education resources to support youth health literacy: A scoping review and environmental scan." Health Education Journal 82.7 (2023): 766-778.). Second, there are many reasons HR may not be adopted, and health professional's resistance is only one – currently this sentence reads as if you are suggesting it is the only reason – please rephrase. 9) P10 I7 As a very minor side note, it's interesting that you haven't used covidence for all of this process from start to finish?
--	--

	10) P11 5 I am not clear that excluding systematic reviews is sensible given they usually provide a rigorous comprehensive review and useful conclusions – I would suggest either including them (and ensuring duplication doesn't occur) or explain in more detail why not 11) P15 5 Suggest rephrasing – there is an assumption that practitioners need to change their practice = perhaps recognising some may already adopt HR is valuable, and secondly, to show balance overall
--	---

VERSION 1 – AUTHOR RESPONSE

A) Reviewer: 1
Dr. Ellie Grossman, Cambridge Health Alliance

Comments	Responses	Page number (#)
[x] Major Essential Revisions:  • The Introduction section on 'HR efficacy' would be more compelling if the authors added quantitative data supporting effectiveness and/or efficacy of harm-reduction programs. 	Comment addressed and resolved. The revised version of the protocol presents additional quantitative data supporting the effectiveness of harm reduction in cannabis use. Some of the added references are the following: Fischer B, Jones W, Shuper P, et al. 12-month follow-up of an exploratory 'brief intervention' for high-frequency cannabis users among Canadian university students. Subst Abuse Treat Prev Policy 2012;7(1):15. doi: 10.1186/1747-597X-7-15. Sherman BJ, Sofis MJ, Borodovsky JT, Gray KM, McRae-Clark AL, Budney AJ. Evaluating cannabis use risk reduction as an alternative clinical outcome for cannabis use disorder. Psychol Addict Behav.	5-6

	2022;36(5):505. doi: 10.1037/adb0000760.	
Minor Essential Revisions: [x] • The authors should include the specific dates for their literature search.	Comment addressed and resolved. All search strategies were executed on October 10th, 2022 and this information has been added to the abstract and main text.	1-9-10
[x] • It is confusing (and contradicts their thesis) that the authors say the rate of cannabis use in Canada and Quebec is declining (Introduction, first paragraph) – because it seems to contradict data from other sources and parts of the world, and it weakens the impetus for the authors’ study. I would suggest the authors omit this sentence.	Comment addressed and resolved. The sentence has been omitted.	3
[x] • One aim of the authors’ study is to provide evidence they can use in improving systems of care for adolescents and young adults. In the Introduction section, some of their rationale and background focuses on this population as well (e.g. the ‘HR efficacy’ section). However, their scoping review does not restrict inclusion based on targeted population age – which makes for a mismatch between background/rationale provided and the actual study planned. The manuscript may hang together better if the authors focus on a non-age-restricted rationale for their study, and then mention a focus on providers serving adolescents and young adults as an exploratory substudy (if there is sufficient existing literature found).	Comment addressed and resolved. The revised version of the protocol focuses more on HR in cannabis use in general, without restricting it to a specific age range. However, in the “effectiveness” section, we presented the effectiveness of HR across various populations (i.e., youth, adults, etc.).	3-4-5-6
Other Comments: [x] • A strength of this study is its use of the term ‘healthcare providers’; I would expect that the authors will find studies involving a range of different types of health-care workers. These professional differences may need to be taken into account in the analytic plan or way of presenting results. The authors do a nice job of explaining what a scoping review is (first paragraph of	That is correct, we did find studies involving a range of different types of healthcare workers. In the scoping review (not the protocol), the results will be presented by taking into account the different “types” of healthcare providers that we found in our research.	X

Methods/Analysis section). Table 1 is also very useful.		
---	--	--

A) Reviewer: 2
Dr. Nancy Sohler, City University of New York

Comments	Responses	Page number (#)
[x] 1. page 6: Efficacy and effectiveness are both used, although I think the studies are effectiveness studies.	Comment addressed and resolved. We opted the term "effectiveness".	5
[x] 2. The introductory sections seem to focus on young people, but this isn't highlighted in the main research question(s) and/or the methods.	Comment addressed and resolved. The revised version of the protocol focuses more on HR in cannabis use in general, without restricting it to a specific age range. However, in the "effectiveness" section, we presented the effectiveness of HR across various populations (i.e., youth, adults, etc.).	3-4-5-6
[x] 3. I assume the years for the journal index searches are the same as for the database searches--is that correct? Also a list of journals to be included in this process probably should be identified ahead of time.	Comment addressed and resolved. The search strategy was executed on all databases on October 10th, 2022, and the included studies will be those published between 1990 and October 10th, 2022.	1-9-10
[x] 4. I'm confused about the use of inter-rater reliability in the article selection process. It seems the two reviewers will discuss disagreements and then reliability will be calculated. Wouldn't discussion lead to a consensus?	Inter-rater agreement will be calculated based on the initial number of conflicts between the two reviewers, and thus, before reaching consensus. By the end of the selection process, a consensus will have been reached for all the selected articles. Comment addressed in the main document.	10

[x] 5. It appears that one person will be collecting all of the data after the studies are selected (with some validation from another person). I believe there is potential for bias and therefore >1 person could be included.	The data extraction sheet template will be validated by all the authors. By doing so, the data that needs to be extracted will be clear and well-defined. Following data extraction, the research supervisor will review and validate all data extraction sheets. Two reviewers will be involved in the study selection stage to limit any source of bias. However, engaging two persons to independently extract data would be difficult to achieve due to limited resources.	X

B) Reviewer: 3
Ms. Hannah Walsh, King's College London

Comments	Responses	Page number (#)
[x] 1) P4 line 24 sentence Other research suggests cannabis use transition to dependence is significant – not sure what 'limited addictive power' refers to but suggest rephrasing, also references cited there are old – suggest finding more recent references – e.g.	Comment addressed and resolved. The revised version of the protocol mentions the following: "Cannabis use does not necessarily lead to dependence due to several factors such as infrequency in use, consumption of cannabis with low rates of δ-9-tetrahydrocannabinol (THC), availability, legality, and social acceptability (9, 11-15). The lifetime cumulative probability of transitioning from use to dependence was found to be the lowest for people who use cannabis (PWUC) (8.9%) compared to other substances such as nicotine (67.5%), alcohol (22.7%), and cocaine (20.9%) (15). Even with long-term exposure to cannabis, PWUC do not necessarily develop severe problems or a cannabis-dependence (5, 15, 16)."	3
Lopez-Quintero, Catalina, et al. "Probability and predictors of transition from first use to dependence on nicotine, alcohol, cannabis, and cocaine: Results of the National Epidemiologic Survey on Alcohol and Related Conditions (NESARC)." Drug and alcohol dependence 115.1-2 (2011): 120-130. Englund, Amir, et al. "Can we make cannabis safer?." The Lancet Psychiatry 4.8 (2017): 643-648.	Comment addressed and resolved. The mentioned references have been added.	3
[x] 2) Suggest rephrasing p4 line 28 – to discuss 'health and psychosocial harms and consequences'?	Comment addressed and resolved. The revised version of the protocol mentions the following: "These harms may include a decreased academic or professional performance, a cognitive impairment, a deterioration of mental health (e.g., development of psychosis or depression), an increased	3

	occurrence of risky behaviors (e.g., cannabis-impaired driving), etc. (3-5, 12, 14, 18, 20-22)”	
[x] As a general point, it might be useful to situate the issue more clearly within the context of harms and widespread use; since one of the issues with cannabis is that it is widely used and has negative and harmful consequences for a proportion of users – not all - but that vulnerability to harmful consequences may not be obvious prior to use, or may not prompt different choices about use. Some of the consequences can be very significant (e.g. psychosis) – so the frequency might be low but the impact is huge, hence harm reduction has an important role to play.	Comment addressed and resolved. The revised version of the protocol mentions the following: “However, PWUC intensively or at-risk populations (i.e., pregnant persons, people presenting respiratory problems, a mental health comorbidity, or having an early onset of continued tobacco smoking) may develop a cannabis use disorder or experience harm at several levels (5, 12, 16-19). These harms may include a decreased academic or professional performance, a cognitive impairment, a deterioration of mental health (e.g., development of psychosis or depression), an increased occurrence of risky behaviors (e.g., cannabis-impaired driving), etc. (3-5, 12, 14, 18, 20-22). It is important to specify that a severe cannabis-induced mental health condition (e.g., psychosis) might occur among only 2% of PWUC (5). Despite the minimal probability of leading to potentially serious adverse consequences, the high prevalence of cannabis use as well as the potential harms that might be experienced, make it essential to implement effective intervention programs among PWUC (7, 12, 15, 17).”	3
[x] Not least, co-administered tobacco use is a significant factor and may add to dependence risk.	Comment addressed and resolved, as indicated in the previous comment response: “However, PWUC intensively or at-risk populations (i.e., pregnant persons, people presenting respiratory problems, a mental health comorbidity, having an early onset of continued tobacco use or concurrently smoking tobacco and cannabis) may develop a cannabis use disorder or experience harm at several levels (5, 12, 16-19).”	3

[x] In this section, the literature cited is quite old – cannabis potency AND research has developed a lot in last ten yrs so it is important to reflect this	Comment addressed and resolved. The references have been updated and include the following studies: Fischer B, Russell C, Sabioni P, Van Den Brink W, Le Foll B, Hall W, et al. Lower-risk cannabis use guidelines: a comprehensive update of evidence and recommendations. Am J Public Health. 2017;107(8):e1-e12. Englund A, Freeman TP, Murray RM, McGuire P. Can we make cannabis safer? Lancet Psychiatry. 2017;4(8):643-8. doi: 10.1016/S2215-0366(17)30075-5. Lopez-Quintero C, de los Cobos JP, Hasin DS, Okuda M, Wang S, Grant BF, et al. Probability and predictors of transition from first use to dependence on nicotine, alcohol, cannabis, and cocaine: Results of the National Epidemiologic Survey on Alcohol and Related Conditions (NESARC). Drug Alcohol Depend. 2011;115(1-2):120-30. doi: 10.1016/j.drugalcdep.2010.11.004. Hindocha C, Shaban ND, Freeman TP, Das RK, Gale G, Schafer G, et al. Associations between cigarette smoking and cannabis dependence: a longitudinal study of young cannabis users in the United Kingdom. Drug Alcohol Depend. 2015;148:165-71. doi: 10.1016/j.drugalcdep.2015.01.004. Lemyre A, Poliakova N, Bélanger RE. The relationship between tobacco and cannabis use: a review. Subst Use	3-4-5

	Misuse. 2019;54(1):130-45. doi: 10.1080/10826084.2018.1512623. Winer JM, Yule AM, Hadland SE, Bagley SM. Addressing adolescent substance use with a public health prevention framework: the case for harm reduction. Ann Med. 2022;54(1):2123-36. doi: 10.1080/07853890.2022.2104922.	
[x] 3) P4 line 38 Not all programs will have been based on abstinence – important to qualify language – eg. War on drugs etc has influenced aims of many prevention/treatment programs. Suggest including wider, recent references here.	Comment addressed and resolved. The sentence has been modified and the revised version mentions the following: “To address this reality, prevention and treatment programs based on the abstinence approach (i.e., total elimination of cannabis use), have been widely implemented (11, 13). The emergence of these programs has been also influenced by policies such as the “War on Drugs” (23). The abstinence-based model forms the basis of many programs developed to prevent or treat problematic cannabis use [...]”	4
For example, the paper below is a key text: Fischer, Benedikt, et al. "Lower-Risk Cannabis Use Guidelines (LRCUG) for reducing health harms from non-medical cannabis use: A comprehensive evidence and recommendations update." International journal of drug policy 99 (2022): 103381.	Comment addressed and resolved. The mentioned reference has been included and added to different sections (“Cannabis use”; “Harm reduction - Description”; “Acceptability of harm reduction”).	3 → 7
Overall, it might be easier to make the argument for this review by describing the literature focussing on harm reduction relating to cannabis only, rather than harm reduction per se. Harm reduction across all substance use is very broad, and as described already cannabis is a very different substance to heroin, cocaine use, which have driven the impetus for harm reduction approaches thus far.	Comment addressed and resolved. The revised version of the protocol focuses more on harm reduction in cannabis use specifically.	3 → 7

Therefore, it may be more useful to review extant literature on young adult harm reduction, and cannabis harm reduction as two foci.	The revised version of the protocol focuses more on harm reduction in cannabis use specifically. However, we did not restrict our introduction to a specific age-group because our research objective does not only tackle healthcare providers working among youth. This being said, we presented the effectiveness of harm reduction across various populations (i.e., youth, adults, etc.).	3 → 7
e.g. : Kruger, Jessica S., Daniel Kruger, and R. Lorraine Collins. "Knowledge and practice of harm reduction strategies among people who report frequent cannabis use." Health promotion practice 22.1 (2021): 24-30.	The mentioned reference has been included and added to different sections ("Abstinence approach"; "Acceptability of harm reduction").	7
Sherman, Brian J., et al. "Evaluating cannabis use risk reduction as an alternative clinical outcome for cannabis use disorder." Psychology of Addictive Behaviors 36.5 (2022): 505.	The mentioned reference has been included and added to the following section : "Harm reduction - Effectiveness".	5-6
Chatham, Carrington L. Harm Reduction: Social Workers' Attitudes toward Marijuana Use in Substance Abuse Treatment . Diss. Capella University, 2021.	We were not able to access the full thesis. However, the research questions are not directly related to our objective (i.e., facilitators and obstacles to the adoption of HR in cannabis use). The research questions were the following and tackled HR methods in general and the use of cannabis as an HR strategy: 1. What are social workers' attitudes in the substance abuse treatment field towards using harm reduction methods in general?	X

	2. What are social workers' attitudes in the substance abuse treatment field toward using marijuana as a harm reduction method? 3. What are social workers' attitudes in the substance abuse treatment field about the changing legal status of marijuana in the United States? 4. What are social workers' attitudes in the substance abuse treatment field toward expanding clients' options to include abstinence-only and harm reduction strategies? 5. Study participants can develop what action plan to broaden treatment options for substance abusing clients that include all evidence-based treatments?	
Winer, James Michael, et al. "Addressing adolescent substance use with a public health prevention framework: the case for harm reduction." Annals of medicine 54.1 (2022): 2123-2136.	The mentioned reference has been included and added to different sections ("Cannabis use"; "Harm reduction – Description").	3-4
[x] 4) L54 I'm not sure this is a strong argument – again it's difficult to make general statements about 'abstinence programs' – there may be plenty which do reference peer pressure and social context. The third argument is stronger.	Comment addressed and resolved. The sentence was rephrased. The revised version mentions the following: "Second, the abstinence approach tends to focus more on the negative consequences of use through strategies that evoke fear without necessarily taking into consideration the social context of cannabis use".	4

☒ 5) P5 l26 – clarify that authors are describing the argument that HR is based upon – it reads like authors’ own opinion	Comment addressed and resolved. The revised version of the protocol mentions the following: “The effectiveness of HR among youth who use cannabis has led the University Institute on Addictions (Institut universitaire sur les dépendances) in Quebec to recommend it as an intervention modality among this clientele (36).”	5
☒ 6) P6 l8 – for ‘psychiatric disabilities’ I assume you mean ‘mental health conditions’ ?	Comment addressed and resolved. Yes, we mean “mental health conditions”.	5
☒ 7) P6 l36 I suggest authors clarify which healthcare providers you are referring to (this is a huge group – are you focussing on substance use providers? Mental health?) – the statement that HR is ‘controversial’ – this seems a little strong. I suggest instead something like ‘not widely accepted’.	Comment addressed and resolved. Changes have been made to the initial sentence and the revised version mentions the following: “Despite its proven effectiveness, the acceptability of HR among health and social services practitioners remains limited, and various factors facilitate or hinder its adoption”.	7
☒ 8) P7 l 29 This statement needs qualifying – to justify that it is ‘under-implemented’ would require substantial references which have examined the adoption of HR approaches, for example at a policy level; (e.g. Emily J., et al. "Canadian cannabis education resources to support youth health literacy: A scoping review and environmental scan." Health Education Journal 82.7 (2023): 766-778.). Second, there are many reasons HR may not be adopted, and health professional’s resistance is only one – currently this sentence reads as if you are suggesting it is the only reason – please rephrase.	Comment addressed and resolved and the suggested reference has been added. The revised version of the protocol mentions the following: “Despite its proven effectiveness, the acceptability of HR among health and social services practitioners remains limited, and various factors facilitate or hinder its adoption (21, 40-42)”.	7

[x] 9) P10 I7 As a very minor side note, it's interesting that you haven't used covidence for all of this process from start to finish?	Covidence was used for the whole "study selection" stage. It is noted in the third stage of the methodology.	10
[x] 10) P11 I5 I am not clear that excluding systematic reviews is sensible given they usually provide a rigorous comprehensive review and useful conclusions – I would suggest either including them (and ensuring duplication doesn't occur) or explain in more detail why not	Before excluding any potentially relevant systematic review, we will be scanning its reference list as well as its list of included studies. By doing so, we might find relevant empirical studies that are related to our study's objective and that were not initially captured by our search strategy. These additional studies (i.e., that are directly related to our study's objective), will be included. Our search strategy might have already captured the studies included in a systematic review. By excluding systematic reviews and including empirical studies related to our objective, we will avoid duplication and ensure equal representation of the selected papers.	10
[x] 11) P15 I5 Suggest rephrasing – there is an assumption that practitioners need to change their practice = perhaps recognising some may already adopt HR is valuable, and secondly, to show balance overall	Comment addressed and resolved. The revised version of the protocol mentions the following: "After completing the scoping review, we will be able to conduct a future study aiming to implement a knowledge translation plan among practitioners working with youth in Quebec to enhance and expand their adoption of HR in cannabis use".	14

VERSION 2 – REVIEW

REVIEWER	Grossman, Ellie Cambridge Health Alliance
REVIEW RETURNED	16-Mar-2024

GENERAL COMMENTS	The authors have answered the prior version's comments sufficiently and completely.
---

REVIEWER	Walsh, Hannah King's College London, Department of Addictions
-----------------	--

REVIEW RETURNED	29-Feb-2024
-------------

GENERAL COMMENTS	Thank you for the opportunity to review this revision. The authors have addressed the issues raised by the reviewers and I am satisfied with the responses, I have a minor suggestion to make as listed below: Introduction: Cannabis use: Re paragraph starting 'cannabis use does not necessarily lead to dependence ...' this line seems unnecessary to me - and the same point could be made more simply by only including the second line 'The lifetime cumulative probability...'. I think this argument is sufficient and stronger than the argument in the first sentence.
--

VERSION 2 – AUTHOR RESPONSE

A) Comments of reviewer 3:

Introduction: Cannabis use: Re paragraph starting 'cannabis use does not necessarily lead to dependence ...' this line seems unnecessary to me - and the same point could be made more simply by only including the second line 'The lifetime cumulative probability...'. I think this argument is sufficient and stronger than the argument in the first sentence.	Comment addressed and resolved. The revised version of the protocol mentions the following: “The lifetime cumulative probability of transitioning from use to dependence was found to be the lowest for people who use cannabis (PWUC) (8.9%) compared to other substances such as nicotine (67.5%), alcohol (22.7%), and cocaine (20.9%); this can be due to several factors such as infrequency in use, consumption of cannabis with low rates of δ-9-tetrahydrocannabinol (THC), availability, legality, and social acceptability (9, 11-15)”.	3
---	--	----------